# Deep convolution neural network model for automatic risk assessment of patients with non-metastatic nasopharyngeal carcinoma

**Richard Du**[1]                                                                               DU94@HKU.HK
[1] *Department of Diagnostic Radiology, Li Ka Shing Faculty of Medicine, The University of Hong Kong, Hong Kong SAR*

**Peng Cao**[1]                                                                              CAOPENG1@HKU.HK

**Lujun Han**[2]                                                                          HANLJ@SYSUCC.ORG.CN
[2] *Sun Yat-Sen University Cancer Center, Guangzhou, China*

**Qiyong Ai**[3]                                                                         AQY0621@CUHK.EDU.HK
[3] *Department of Imaging and Interventional Radiology, Faculty of Medicine, The Chinese University of Hong Kong, Hong Kong SAR*

**Ann D. King**[3]                                                                    KING2015@CUHK.EDU.HK

**Varut Vardhanabhuti**[1]                                                                    VARV@HKU.HK

## 1. Introduction

Nasopharyngeal Carcinoma (NPC) is a malignant tumour that develops in the nasopharynx. NPC is endemic in the south-east Asia with approximately a third of the world incidences (Tang et al., 2016). With the advent of intensity-modulated radiotherapy, better sparing of adjacent organs at risk and excellent locoregional control are being achieved. Consequently, this had led to pretreatment clinical staging classification to be less prognostic of outcomes such as recurrence after treatment (Liu et al., 2017). Alternative pretreatment strategies for prognosis of NPC after treatment are needed to provide better risk stratification for NPC. In this study, we attempt to develop an automatic deep convolution neural network model to predict 3-year disease progression.

## 2. Methods

### 2.1. Data set

596 non-metastatic NPC patients were retrospectively obtained from four independent centres in Hong Kong and China. Head and neck contrast enhanced T1-weighted (T1C) and T2-weighted (T2) MRI scan along with routine pretreatment TNM staging classification and 3-year progression-free survival status were retrieved for the study. The 3-year disease progression rate was found to be between 10-20% across the cohorts. All scans were resampled to 1x1x6cm resolution. For development of the model, 450 patients from three centres were allocated for the training set, and 146 from one other centre was allocated for the testing set. The primary NPC tumours of a subset of the 285 patients from training set

and 30 patients from testing set were manually delineated by a board-certified radiologist. All manual segmentation was performed on the T1C scans.

## 2.2. Proposed model and training

The proposed model consists of two separate networks which perform segmentation of the primary NPC tumour and then uses segmentation for classification of 3-year disease progression. For segmentation, inspired by the VNet architecture (Milletari et al., 2016), we modified the architecture to provide two separate encoding routes for encoding the T1C and T2 scans. The resulting encoded latent features from the two encoders are combined for the segmentation decoder. Additionally, a classification encoder was added to classify the overall and T stage of the tumour. The intuition behind the classification route is to force the network to encode features related to the tumour. The proposed network architecture, named as V2NetCls, are shown in Figure 1. For training, 15% of the subset of patients with segmentation was randomly split for validation (n = 43) and the rest used for training (n = 243). Adam optimiser with a learning rate of 1e-4 and batch of 2 was used. Dice loss was used as it known to be superior in class imbalanced situation (Milletari et al., 2016). For baseline, a VNet trained with T1C scan only and the V2Net without classification was also trained. Once the segmentation network is trained, the output was used to crop a 64x64x12 bounding box of the segmented tumour on T1C and T2 scan for classification of 3-year PFS. ResNet18 was used as the network to classifier 3-year disease progression (He et al., 2015). The input to the network is a 64x64x12 volume with three channels corresponding to the T1C, T2 and the segmentation mask. Similarly, an 85/15 split was used for training (n = 384) and validation (n = 68). Weighted binary cross-entropy loss was used for training.

## 3. Results

The segmentation performance of the proposed and baseline networks is given in Table 1. The proposed V2NetCls achieved the best performance compared to the two baseline networks, hence was subsequently used for the classification of 3-year disease progression. The classification network achieved an AUC = 0.828 (sensitivity = 0.833; specificity=0.887) in the validation set but was unable to generalise to the testing set (AUC = 0.689; sensitivity = 0.685; specificity = 0.716)

Table 1: Segmentation performance of the experimented networks. IQR, interquartile-range

| Model | Data set | Median Dice (IQR) |
| --- | --- | --- |
| VNet T1C | Validation set (n = 43) | 0.558 (0.397 - 0.646) |
| | Testing set (n = 30) | 0.437 (0.329 - 0.565) |
| V2Net T1C+T2 | Validation set (n = 43) | 0.589 (0.455 - 0.675) |
| | Testing set (n = 30) | 0.559 (0.429 - 0.656) |
| V2NetCls T1C+T2 | Validation set (n = 43) | 0.612 (0.456 - 0.693) |
| | Testing set (n = 30) | 0.585 (0.480 - 0.681) |

## 4. Discussion

Despite the low multicentre performance, our preliminary results show that deep learning may offer prognostication of disease progression of NPC patients after treatment. Despite sub-par segmentation performance, the segmentation was able to localise the tumour and act as prior knowledge for the classification. More samples number may be needed to increase the segmentation performance as seen in (Lin et al., 2019). The classification performance was comparable to a previous NPC model based on hand-crafted radiomics features which achieved an AUC = 0.78 in a internal test set (Zhang et al., 2017). One advantage of our model is that it does not require manual segmentation of the region of interest, hence reducing clinician's burden. However further development in generalising multicentre data set are needed before clinical application of deep learning models in assessment of NPC.

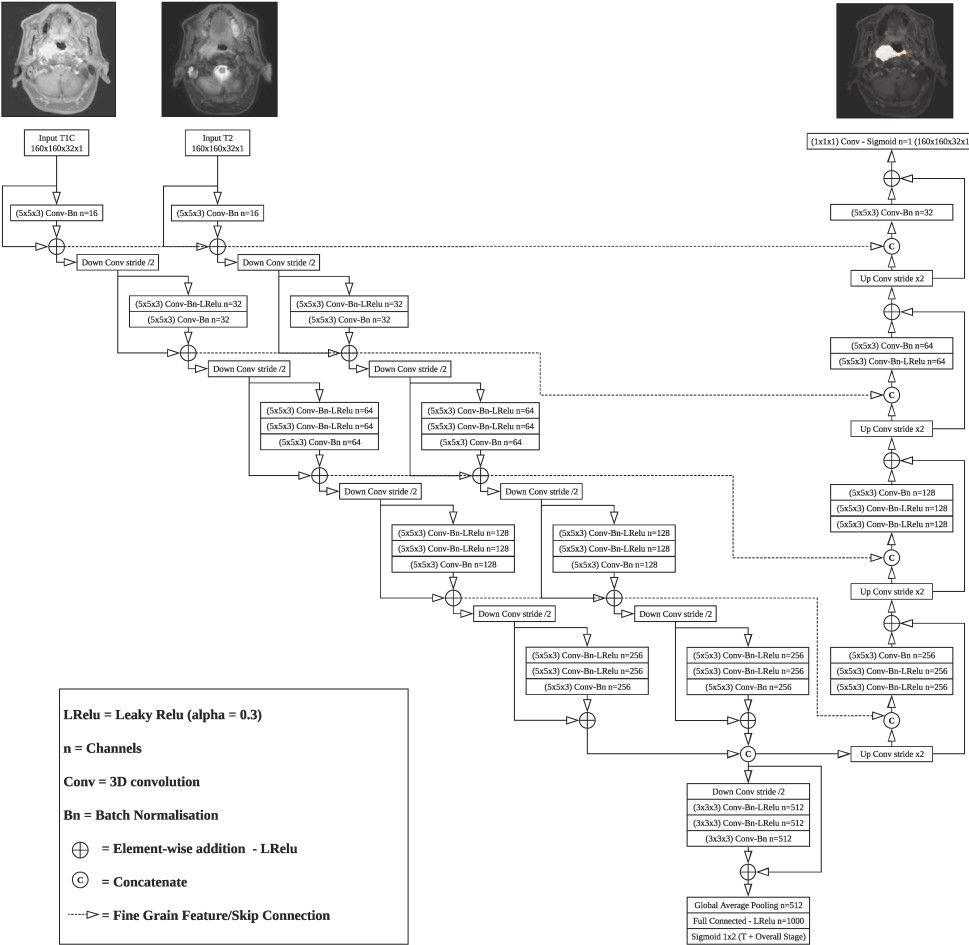

Figure 1: Architecture of the proposed V2NetCls for segmentation of primary NPC tumour

## Acknowledgments

RD is supported by a postgraduate fellowship from the Lee Shau Kee Foundation.

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
