# OpenReview forum: "Deep convolution neural network model for automatic risk assessment of patients with non-metastatic nasopharyngeal carcinoma"
_MIDL.io/2019/Conference/Abstract — MIDL Abstract 2019_

### Official Review · AnonReviewer1 · 2019-04-28
**improve results, comparative study, "sub-par" performance, missing evaluation of disease progression rates**

**Rating:** 2
**Confidence:** 3

**Review:**

Deep Convolution Neural Network Model for Automatic Risk Assessment of Patients with Non-metastatic Nasopharyngeal Carcinoma

This paper uses two VNets on T1 and T2 images to, first, segment nasopharynx tumors, and second, predict a 3-yr disease progression rate. The method uses existing VNets. The results shows "sub-par" segmentation performance (Table 1), and lack a report of disease progression ("unable to generalize on the testing set").
The submission in its current form may benefit from refined results, convincing with a comparative study. The shown results currently shows only a comparison with its own variants, the said "sub-par" scores of 43-58% Dice should be related to what is achieved on pharynx tumor segmentation. Disease rates should also be related. As is, it may be difficult to judge on the significance of the proposed method.

---

### Official Review · AnonReviewer2 · 2019-05-01
**The method and preliminary results are clearly presented, looking into generalisation to multicentric data is a nice addition.**

**Rating:** 4
**Confidence:** 2

**Review:**

The abstract report preliminary investigations of DL architectures for prediction of 3-year disease progression of NPC patients from T1C and T2 head & neck MRI. The approach solves an auxiliary segmentation task using an encoder-decoder (+ skip connections) architecture, whereby the bottleneck encoding is also encouraged to enable accurate classification via an additional route. Then both input scans and segmentation mask are passed to a ResNet trained specifically for classification (prediction of disease progression).

The approach, preliminary results and investigations (including the multicentric twist) are clearly and soundly exposed and the developments should be interesting.

---

### Decision · Program_Chairs · 2019-05-06
**Acceptance Decision**

Accept